# One Health Surveillance of Antimicrobial Use and Resistance: Challenges and Successes of Implementing Surveillance Programs in Sri Lanka

**DOI:** 10.3390/antibiotics12030446

**Published:** 2023-02-23

**Authors:** Sujeewa Ariyawansa, Kuruwitage N. Gunawardana, Muditha M. Hapudeniya, Nimal J. Manelgamage, Chinthana R. Karunarathne, Roshan P. Madalagama, Kamalika H. Ubeyratne, Darshana Wickramasinghe, Hein M. Tun, Peng Wu, Tommy T. Y. Lam, Olivia S. K. Chan

**Affiliations:** 1National Aquatic Resources Research and Development Agency, Crow Island, Colombo 01500, Sri Lanka; 2Ministry of Health, Medical Research Institute, Colombo 00800, Sri Lanka; 3Ministry of Health, 385, Suwasiripaya, Colombo 01000, Sri Lanka; 4Department of Animal Production and Health, No. 13, Getambe, Peradeniya, Kandy 20400, Sri Lanka; 5Department of Animal Production and Health, Veterinary Investigation Centre, Court Road, Wariyapola 60400, Sri Lanka; 6Veterinary Research Institute, Peradeniya 00902, Sri Lanka; 7Ministry of Health Sri Lanka, District General Hospital, Hambantota 81000, Sri Lanka; 8The Jockey Club School of Public Health and Primary Care, Faculty of Medicine, The Chinese University of Hong Kong, Hong Kong SAR, China; 9The School of Public Health, Li Ka Shing Faculty of Medicine, The University of Hong Kong, 7 Sassoon Road, Pokfulam, Hong Kong SAR, China

**Keywords:** Sri Lanka, One Health, antimicrobial resistance, antimicrobial use/consumption, surveillance, implementation

## Abstract

Background: Sri Lanka is a low-income country, as defined by the World Bank. The country suffered further economic downturn during the COVID-19 pandemic. This situation adversely affected the prioritization of policies and programs around healthcare and public health. In particular, inflation, fuel prices, and shortage of food supplies increased struggles to implement antimicrobial resistance (AMR) programs. However, in the long run, it is crucial to gather data and evidence to plan AMR policies and track interventions. (1) Aim: To establish and reiterate the importance of prioritizing AMR programs in the One Health framework, the Fleming Fellows collected and studied antimicrobial use/consumption (AMU/AMC) and resistance (AMR) in humans, food-producing animals, and the environment. (2) Methods: A systematic and cross-sectional study was conducted between 2019 and 2021. By way of coordinating an AMU/AMC and AMR prevalence study across six agencies from human health and food-producing animal sectors, the authors established a field epidemiology study, laboratory testing, and data processing at their institutions. AMU/AMC patterns were surveyed using questionnaires and interviews, while AMR samples were collected for antibiotic susceptibility tests and genomic tests. Samples were tested for phenotypic and genotypic resistance. (3) Results: In human samples, resistance was highest to beta-lactam antibiotics. In non-human samples, resistance was highest to erythromycin, a highest-priority, critically important antibiotic defined by the World Health Organization. From government records, tylosin was sold the most in the food-producing animal sector. (4) Conclusions: Sri Lanka AMU and AMR trends in human and non-human sectors can be ascertained by a One Health framework. Further coordinated, consistent, and sustainable planning is feasible, and can help implement an AMU/AMR surveillance system in Sri Lanka.

## 1. Introduction

Antimicrobial resistance (AMR) is an identified threat in the global public health system [1,2,3]. Importantly, antimicrobial resistance (AMR) has rapidly emerged in humans in the form of pan- and multi-drug resistant bacteria, which are endemic in various countries in the world [4]. Recent research by the World Bank indicates that AMR will elevate poverty rates and disproportionally affect low-income countries and vulnerable economies, such as that in Sri Lanka [5] (“Drug-Resistant Infections: A Threat to Our Economic Future (Vol. 2): Final Report (English); https://documents.worldbank.org, accessed on 10 February 2023). Sri Lanka relies on aquaculture and terrestrial farming for local food supply and commercial export. The country’s healthcare system is in need of resources and capacity building. The country as a whole faces multiple problems, with antimicrobial use and resistance issues in both the human and animal sectors, sliding from policy, economic or political priority.

Antimicrobial usage in humans, livestock, and aquaculture, collectively, have been identified as a good indicator of the risk of antimicrobial resistance [6,7]. While it is true that antimicrobial use (AMU) and AMR surveillance are fundamental data to monitor baselines and trends, implementation of surveillance is challenging because it requires collective governance and infrastructure prioritization among healthcare delivery systems, laboratory capacities, farming, and infectious disease control entities. Despite challenges, surveillance of AMU and AMR in the One Health framework can help prioritize resource allocation and policy prioritization [8,9].

Following Sri Lanka’s AMR National Action Plan established in 2016, Sri Lanka formulated surveillance guidelines and platforms established by the World Health organization (WHO) and the World Organization for Animal Health (WOAH). It established tripartite collaboration with the World Health Organization (WHO) and its Global Antimicrobial Resistance Surveillance System (GLASS), the World Organization for Animal Health (WOAH), and the Food and Agriculture Organization (FAO). Since then, human and animal antimicrobial resistance and consumption have been overseen by the Ministry of Health and the Department of Animal Production and Health. In the human sector, studies described opportunities to improve antimicrobial use in hospitals [10] and highlighted issues of self-medication among patients [11]. In the animal sector, Sri Lanka faces pressure to supply more food for a growing population, with risk of increasing antimicrobial use in food-producing animal sectors [12]. In particular, foodborne pathogen, such as *Escherichia coli* (*E. coli*), are highly transmissible via food source contamination [13,14]. *E. coli* in poultry can be pathogenic and has exhibited multi-drug resistance [15], zoonotic [16], and soil contamination [17] characteristics. The pathogenic strain is also producing an emerging lineage of resistance, including CTX-M-15 enzyme and fluoroquinolone resistance [18].

All of human, food, and animal sectors are experiencing challenges that require collaboration to plan and prioritize intervention and stewardship programs. However, a comprehensive study on antimicrobial usage (AMU) and AMR in human and animal sectors was not conducted in Sri Lanka. With the intention to extend the current surveillance effort on AMU and antimicrobial resistance (AMR) under the Fleming Fellowship program, this group of professionals in the offices of governments, laboratories, and hospitals collaborated to implement an AMU and AMR surveillance network in Sri Lanka across human, food-producing, animal farming, and public health sectors. This study investigated three main areas: (1) *Escherichia coli (E.coli)* antimicrobial susceptibility pattern across the human, poultry (broiler), and aquatic (prone) sectors; (2) genetic variations in resistant isolates; and (3) antimicrobial use and consumption in commercial broilers and aquatic (prone) sectors. This article describes cross-sectional data of AMU and AMR in a One Health framework, as well as multivariate analysis of variables in human healthcare and food-producing animal sectors in Sri Lanka.

## 2. Results

### 2.1. Overview of AMU and AMR in Non-Human Sectors

Assessing antimicrobial use and consumption in the non-human sector, tylosin contributed to 37% of the use, and was the most commonly used antimicrobial in poultry farms. AMU calculation was adopted from Umair M. et al. [19]. There was 49,117,000 mg of active substance (49.11 kg) used on an estimated biomass of 256,477.5 kg Population Correction Unit (PCU) in poultry farming. Antimicrobial usage in poultry was calculated to be 191.51 mg/PCU in the two divisional secretariats, collectively. Altogether, 187 kg of different antimicrobials was used among poultry farms, in which sulfaquinoxalin, amoxicillin, enrofloxacin, and tylosin were identified as the main antimicrobials used. Enrofloxacin was the most common antimicrobial prescribed for poultry farmers by government veterinarians.

In terms of resistant isolates in poultry samples, over 90% of resistance was identified against erythromycin, tetracycline, and beta-lactams. In aquaculture, the resistance to erythromycin and gentamicin were observed at 90.2% and 45.1%, respectively. In terms of resistant isolates in human samples, *E. coli* isolates from human urine samples depicted resistance to multiple antibiotics. In descending order of frequencies, these isolates were resistant to beta-lactams (66.5%), amoxicillin–clavulanic acid combination (42%), quinolones (35%), tetracyclines (30.5%), and sulfa-trimethoprim (25%). Altogether, 35.5% of isolates from the human, poultry, and aquaculture sectors demonstrated resistance to extended-spectrum beta-lactamase (ESBL), and 25.5% depicted the presence of CTX-M-15 beta-lactamase genes.

### 2.2. Antimicrobial Use in Farms

#### 2.2.1. By Wholesale

Some of the main uses identified by wholesale distribution included tylosin (69.89 mg/PCU), amoxycillin (27.55 mg/PCU), sulphonamide (23.65 mg/PCU), neomycin (18.81 mg/PCU), oxytetracycline (17.12 mg/PCU), and enrofloxacin (11.20 mg/PCU).

#### 2.2.2. By Retail

Through information from farm shops, antimicrobial use distribution on poultry was examined in two selected locations in the country. Altogether, 187 kg of different antimicrobials were distributed among poultry farmers; as categorical data, tylosin (37%), sulfonamide and sulfa-trimethoprim (15%), amoxicillin (14%), neomycin (10%), tetracycline (9%), and enrofloxacin (6%) were identified as the main antimicrobials distributed. Aminopyrividine was identified as the lowest distributed antimicrobial in the selected two regions. By antimicrobial classes, sulfonamides, beta-lactams, macrolids, fluroquinolones, and pleuromutilin were identified as the most distributed antimicrobials in Kuliyapitiya and Paduwasnuwara divisional secretariats.

#### 2.2.3. By Use/Consumption

Furthermore, the farmers surveyed indicated no poultry antimicrobials were used/added to the feed by the manufacturers during the survey period. In addition, no poultry antimicrobials were issued by veterinarians in Panduwasnuwara, Kuliyapitiya, and Chilaw DSDs during the survey period, although they had been prescribed during the whole project period.

#### 2.2.4. By Prescription

Apart from that, amoxicillin trihydrate was considered as the most prescribed antimicrobial for non-poultry clinical cases by government veterinarians during the study. The most frequently used antimicrobials for therapeutic purposes were amoxicillin, enrofloxacin, tylosin, timilcosin, and toltrazuril. In contrast, a total of 2.64 kg of antimicrobials were prescribed by government veterinary surgeons to be used in livestock, excluding poultry, during the study period. As categorical data, beta-lactams, quinolones, and tetracyclines were identified as the common antimicrobial used in other livestock species, such as cattle, buffaloes, goats, pigs, and sheep, during the study period.

### 2.3. Antimicrobial Resistance in Humans and Animals

#### 2.3.1. AMR in the Human Sector

In *E. coli* isolates from human samples, the highest resistance was found against beta-lactams (133/200, 66.5%), amoxicillin–clavulanic acid combination (84/200, 42.0%), quinolones (84/200, 42.0%), tetracyclines (61/200, 30%), and sulfa-trimethoprim (50/200, 25.5%) (Table 1). There was 4% resistance to carbapenems, and no resistance to colistin. In addition, high minimum inhibitory concentrations (MIC) were observed against cephalosporin (1:64 value) and carbapenems (1:16 value) in isolates from humans (7/200) (Table 2).

#### 2.3.2. AMR in the Non-Human Sector

In poultry, over 90% of isolate resistance was identified to erythromycin (393/401, 98%), tetracyclines (388/401, 97%), and beta-lactam (379/401, 95%). The resistances, in descending order, were 84.5% (339/401) to quinolone, 79.8% (317/401) to trimethoprim, 76.1% (308/401) to sulfa-trimethoprim, 34.7% (139/401) to chloramphenicol, and 1.3% (3/401) to carbapenems. In addition, high MICs were observed in three isolates (3/200) against *amoxicillin and* clavulanic acid (>1:64) in *E. coli* isolates from the poultry and aquaculture sectors.

In aquaculture, 90.2% (138/153) of *E. coli* isolated from shrimp muscle exhibited resistance to erythromycin, 45.1% (69/138) to gentamicin, 40.5% (62/153) to beta-lactams, 20.3% (31/153) to tetracyclines, 14.4% (22/153) to quinolones, and 12.4% (19/153) to carbapenems. Except in nalidixic acid, pond water exhibited higher resistance by percentage compared to shrimps (Table 3). In aquaculture, a large number of isolates showed resistance to erythromycin (81.7% in shrimps and 95.7% in water), gentamicin (45% in shrimps and 45.2% in water), ampicillin (28.3% in shrimps and 48.4% in water), tetracycline (16.7% in shrimps and 22.6% in water), nalidixic acid (16.7% for shrimps and 12.9% for water), and chloramphenicol (1.7% for shrimps and 14.08% for water). It was observed that phenotypic antimicrobial resistance in *E. coli* from environmental samples showed a high frequency of resistance (Table 4). In addition, a gap in the CLSI epidemiological cut-off values is not available for amoxicillin, neomycin, and oxytetracycline for the interpretation of water and shrimp [20]. Higher resistance percentages were observed in multiple antimicrobials in water samples when compared to shrimps.

Altogether, 35.5% (71/200) of isolates from all human, poultry, and aquaculture sectors were resistant to extended-spectrum beta-lactamase (ESBL), 25.5% (51/200) of samples were resistant to CTX M 15, and 3.5% (7/200) to CTX M 14 (Table 4). CTX-M or OXA 48 was identified in 67% of *E. coli* isolates from humans, poultry, and aquaculture (Table 5). Among genotypic antimicrobial resistances, carbapenem resistance was observed in 3% (6/200) of *E. coli* isolates.

## 3. Discussion

This study aimed to provide a comprehensive baseline study of antimicrobial resistance and antimicrobial usage in Sri Lanka within the One Health framework of human, poultry, and aquaculture sectors. This study was conducted in locations where the highest poultry (Paduwasnuwara and Kuliyapitiya) and aquaculture farming (Chilaw) take place in the country (shrimp farms are found in northwestern and eastern provinces in the country). This study also described major AMR, AMU, and consumption findings in the limitations of the COVID-19 pandemic and economic downturn. The authors experienced challenges in AMR policy implementation and the importance of feasibility considerations in the context of a low-income country. The COVID-19 travel restriction also posed severe limitations in terms of study design and data collection. This study highlighted where samples and data could be obtained in times of very restricted travel. We focused on antimicrobial usage only in poultry and shrimp aquaculture, as no information was available for fish farming. Our data reflected resistance in human samples, poultry samples, and shrimp and aquaculture environments, but with limited sampling. It is important to plan further studies to produce consistent and systematic longitudinal data. It is important for policy decision-makers, field epidemiologists, stakeholders, and antimicrobial users to prioritize resources and build upon this knowledge for longitudinal studies, field-level surveillance, and prudent antimicrobial-use stewardship programs.

Antimicrobial-use pattern investigation is challenging in various countries, and equally so in poultry farming in Sri Lanka [21,22,23]. By industry, although tylosin is not significant in annual AMU data at VDCA, our data indicated that tylosin is a commonly used antimicrobial in poultry: up to 37% of the total quantity used during the period in the two secretariat divisions (Panduwasnuwara, Kuliyapitiya). Other first-line and critically important antimicrobials in human use described by the World Health Organization, antibiotics such as amoxicillin (14%), sulphonamides and sulfa-trimethoprim combinations (15%), neomycin (10%), oxytetracyline (9%), and enrofloxacin (6%), were commonly used in poultry. Expanding on publications regarding laboratory and epidemiological synergy [24], it is important to consider further AMU study by triangulating the presence of antibiotics in animal muscle and the farming environment by laboratory means, together with sales, and AMU survey data [25].

National wholesale data indicated which antimicrobials were used in the selected three divisional secretariats in Sri Lanka. Annual import quantities of active ingredients varied from 40MT (2016) to 92MT (2018) during the period, and there were 152 veterinary medicinal products (VMPs) containing antimicrobials, imported by 28 importing companies by the end of 2020. The information was based on antimicrobial imports through Sri Lanka Customs, and only legal importations were included. However, the smuggling of antimicrobials through boats in Sri Lanka is not uncommon, especially in fish boats in the northern and eastern parts of the country. In addition, clarification of national wholesale data will help understand the distribution of antimicrobial use in food-producing animal sectors.

The global antimicrobial consumption was estimated to be 63,151 tons in 2010; that will rise by 67% by 2030 [26]. The global average annual consumption of antimicrobials per kg of chicken produced was 148 mg/kg [26]. Even though Sri Lanka uses 50% less than China (318 mg/CPU), current AMU values (191.51 mg/PCU in poultry) in Sri Lanka are still 24 times higher than countries such as Norway (8 mg/PCU) [27].The quantity is underestimated, as some antimicrobials registered for human consumption were used in poultry, and those quantities are not included in national data. A more comprehensive and reflective database will benefit a true estimation of AMU.

A contemporary study estimated the total antimicrobial consumption to be 343.46 million daily defined doses (DDD) in Sri Lanka in 2017 [28]. From clinical experiences, tylosin and quinolone were commonly administered in the selected region, due to the high prevalence of mycoplasmosis and other bacterial infections. An infection prevention and control program regarding *Mycoplasma syanoviae* will help reduce antimicrobial use in commercial poultry farming in Sri Lanka. Authors identified other antimicrobials, such as amoxicillin, sulfamethoxasole, neomycin, tetracycline, and quinolone, as commonly used pharmaceuticals in Sri Lanka’s poultry farming, but such antimicrobial prescription studies, at the farm or veterinarian level, remain challenging.

Our findings in *E. coli* resistance are comparable to those of previous studies in Sri Lanka. Our study identified (100%) of infection associated with UTI samples, which compares to 84% of *E. coli* associated with UTI infections [29]. Our study identified 66.5% of *E. coli* was resistant to beta-lactams, compared to 85% resistance among UTI patients in another study [30]. As an overview, among *E. coli* isolates identified in this study, the resistance percentages observed in descending order were as follows: beta-lactams, quinolone, tetracycline, and sulfa trimethoprim in humans, while the order was observed different in poultry and aquaculture. In poultry, a high frequency of resistance was observed for erythromycin, tetracycline, beta-lactams, quinolone, and sulfa-trimethoprim combinations. On the contrary, a high frequency of resistance was reported for erythromycin, gentamicin, and beta-lactams in the isolates from aquaculture. The consistently high resistance describes a public health issue which indicates an impending need to address AMR in all sectors. All sectors, including human, poultry, and aquaculture sectors, need to collaborate. All policy stakeholders, including political decision-makers, government public health and healthcare offices, professionals who use and prescribe antimicrobials, pharmacies that sell antimicrobials, patients, and the public, need to advocate, formulate, and implement prudent antimicrobial use policies and guidelines.

CTX-M has been identified as the most common resistant determinant which causes ESBL resistance in the country [31]. From our data, CTX-M was associated with ESBL resistance in a subset of isolates. However, individual sector prevalence of CTX-M genes was not evaluated in the study. CTX-M was resistant to beta-lactams, including cephalosporin, which has increased with time in global epidemiology [32]. Similar findings have been reported previously in human *E. coli* isolates in Sri Lanka [30,33]. Moreover, CTX-M was identified as the most common ESBL determinant of *E.coli* in UTI, healthy colonized nasal cavities, perirectal areas and vaginas, previously [30,33,34]. 

Two resistance findings, even at low prevalence, are of concern. First, our finding indicated a low percentage of resistance to ESBL in poultry. Our percentage was found to be lower than PCR tests conducted on phenotypically resistant isolates from commercial broiler chickens. The findings indicated an 18.2% and 0.38% of CTX-M β-lactamases and NDM genes (New Delhi metallo-ß-lactamase), respectively [31]. However, even at a low prevalence, the CTX-M β-lactamases gene is a public health concern because it is considered a rapidly spreading resistant gene among the bacterial species in the family of Enterobacteriaceae [33]. Second, in high-priority critically important antimicrobials in humans, such as carbapenem, our findings indicated low resistance prevalence in humans (4%) and a lower percentage (1.3%) in poultry samples, phenotypically. However, only six isolates were identified as positive genotypically as a collection of all three sectors. A higher resistance percentage was reported (4–8%) in clinical isolates of Enterobacteriaceae spp. previously [35]. Since carbapenem-resistant *E. coli* is associated with difficult and expensive treatment, and likely causes high morbidity and mortality [4], Sri Lanka as a nation needs to prevent and control its spread in the One Health framework to reduce economic, social, and healthcare burdens.

Our study team embraced challenges and implemented feasibility tests in a few areas: first, studying AMU and AMR as a cross-sectional approach requires comprehensive study planning and mobilization of stakeholders with field access. Second, to operate such a study across human and non-human sectors requires significant coordination among public and private offices, as well as prioritization of tasks among entities specialized in human healthcare and food-producing animal sectors. Third, to implement the study in an economically and socially difficult era requires commitment and some financial support. Antimicrobial resistance is linked to antimicrobial consumption and antimicrobial usage in humans and animals. The monitoring of antimicrobial usage and antimicrobial consumption with antimicrobial-resistant patterns will provide vital information on mitigating antimicrobial resistance in humans and animals. Therefore, planning AMR, AMC, and AMU surveys in human and non-human sectors, and aligning the effort at local and national levels, is strongly recommended.

## 4. Materials and Methods

### 4.1. Materials

Samples were collected from three different secretariat areas in North Western Province, Sri Lanka (Figure 1). Sites were based on convenience sampling and where the offices’ outreach staff could travel and had farm connections.

Sample types, sites, and number of samples collected are depicted in Table 5. Findings in terms of sampling focus of antimicrobial resistance, antimicrobial usage, or consumption per sector are summarized (Table 1).

### 4.2. Methods

#### 4.2.1. AMR Investigation in Human and Non-Human Samples

The *E. coli* from urinary tract infections (UTI) was isolated and identified in the regional hospitals. The *E. coli* isolates were identified by colony appearance in the MacConkey agar (Thermo Scientific™ Oxoid™ MacConkey Agar (Dehydrated) Vienna, Austria), Gram stain, and basic biochemical tests, such as TSI, SIM, citrate, and urease tests. The isolates were sent to the Medical Research Institute (MRI) in 2 mL vials of Tryptose soy broth (TSA or Soybean/dehydrated: Oxoid, Thermo Fisher Scientific, Vienna, Austria) casein digest medium, with 40% glycerol, under the cold chain.

Reconfirmation of the *E. coli* was performed at the MRI, and *E. coli* from poultry was collected, isolated, and identified at the Veterinary Research Institute (VRI) from humans and poultry. Similarly, the aquatic samples of *E. coli* isolation were carried out at the National Aquatic Research and Development Agency (NARA). The *E. coli* isolates from three sectors were tested for disk diffusion values at individual laboratories, while MIC values and resistant genes were tested at the MRI. 

Isolates from human *E. coli* isolates were sub-cultured on Tryptic soy agar (TSA or Soybean/dehydrated: Oxoid, Thermo fisher scientific, Vienna, Austria) casein digest medium, and MacConkey/ EMB agar (Eosin Methylene Blue agar (Modified) (Levin) (Dehydrated) Thermo Scientific: Austria, Vienna), and species were confirmed by colony appearance, Gram stain, and basic biochemical tests, such as KIA, TSI, SIM, citrate, urease, MRVP, and indole tests, and further reconfirmations were carried out via the automated identification and AST system (Phoenix 100). The isolates, which passed all three screening tests, were taken as *E. coli* in the study, and included for phenotypic and genotypic antimicrobial susceptibility testing. Antimicrobial susceptibility testing was performed by an automated identification and AST system (Phoenix 100), according to CLSI guidelines (“CLSI document VET01-S2, 2013”, 2013; Janet A. Hindler and Audrey N. Schuetz, 2020). In addition, phenotypic antimicrobial susceptibility on extended-spectrum beta-lactamase, AmpC, and carbapenemase were performed with disk diffusion tests, as described by CLSI guidelines (cefoxitin-resistant strains as AmpC screening positive, modified carbapenem inactivation method, positive strains as carbapenemase-producing, and third generation cephalosporin-resistant strains as extended-spectrum beta-lactamase-producing organisms) [20].

##### Isolates from Poultry

Swabs were enriched in buffered peptone water (BPW) (buffered peptone water/dehydrated: Oxoid, Thermo fisher scientific, Vienna, Austria) casein digest medium, for 24 h at 37 °C, then cultured on eosine-methylene blue (EMB) agar (specific agar for *E. coli*), and incubated for 24 h at 37 °C. Suspected colonies were sub-cultured into 5% sheep blood agar, and incubated for 24 h at 37 °C. A series of biochemical tests, such as oxidase, catalase, TSI, SIM, citrate, urease, MRVP, and indole tests, were conducted to identify *E. coli*. As per cost effectiveness, only one *E. coli* isolate was selected randomly per sample for further testing and stored at −80 °C in Tryptose soy broth (TSA or Soybean/dehydrated: Oxoid, Thermo fisher scientific, Vienna, Austria) casein digest medium, with 20% glycerol. *E. coli* was tested for antibiotics, as listed in Table 1. Species confirmation for *E. coli* was performed using molecular methods, described by Shahein et al., 2021 previously [36]. Antimicrobial susceptibility testing was carried out as described by CSLI guidelines via disk diffusion tests (“CLSI document VET01-S2, 2013”, 2013; “Performance Standard on antimicrobial susceptibility testing, CLSI supplement M 100”, 2019).

##### Isolates from Aquaculture

An antibiotic susceptibility test (AST) was performed on *E. coli* isolates from water (*n* = 93) and shrimp (*n* = 60) samples collected from shrimp ponds associated with North Western Province, Sri Lanka.

A sample of 10 g of shrimp was mixed with 90 mL of maximum recovery diluent (MRD), and a dilution series were prepared using MRD, and added into lauryl sulphate broth (LTB). Samples were incubated at 37 ± 1 °C for 48 ± 2 h for possible gas production in the bottom of Durham tubes, and to observe the opacity of the broth. A loopful from positive LTB tubes was then transferred to two sets of 10 mL of brilliant green bile broth (BGBB) and 10 mL of peptone water, simultaneously, and incubated at 37 ± 1 °C for 48 ± 2 h.

For environmental samples from aquaculture ponds, a dilution series were prepared for the water samples collected from shrimp farms using the MacConkey broth (Thermo Scientific™ Oxoid™ MacConkey Broth (Dehydrated) Vienna, Austria) in 1:9 ratios, and were incubated at 48 ± 2 h at 36 ± 1 °C. Positive results for the MacConkey broth (Thermo Scientific™ Oxoid™ MacConkey Broth (Dehydrated) Vienna, Austria) were identified with positive gas production in the bottom of Durham tubes, and a color change of purple color broth to yellow. A loopful from the positive MacConkey broth (Thermo Scientific™ Oxoid™ MacConkey Broth (Dehydrated) Vienna, Austria) samples was transferred to two sets of 5 mL of BGBB and 10 mL of peptone water, simultaneously, and incubated at 37 ± 1 °C for 48 ± 2 h.

Shrimp and water samples that were positive for BGBB with positive gas production, and positive for Indole reaction by peptone water with the presence of Kovac’s reagent, were then streaked on EMB (eosin-methylene blue agar (Modified) (Levin) (Dehydrated) Thermo Scientific: Austria, Vienna) and incubated at 37 ± 1 °C for 24 ± 2 h. Positive results of EMB agar (eosin-methylene blue agar (Modified) (Levin) (Dehydrated) Thermo Scientific: Austria, Vienna) were identified by the appearance of small, green, metallic, sheen colonies. Isolated metallic, sheen colonies were then streaked on a nutrient agar and incubated at 37 ± 1 °C for 24 ± 2 h. A fresh culture of suspected colonies on the nutrient agar was then subjected to biochemical tests. The TSI agar test and MR-VP tests were performed for suspected colonies: the colonies showed TSI agar results with a yellow slant, yellow butt with positive gas production, and positive MR test and negative VP test colonies were biochemically confirmed as *E. coli*. Biochemically confirmed *E. coli* colonies were then subjected to antimicrobial susceptibility testing, as previously described by CLSI guidelines. (“Performance Standard on antimicrobial susceptibility testing, CLSI supplement M 100”, 2019).

##### Genomics Analysis

The represented samples were collected based on the minimum required sample for the interpretation. The isolates were sub-cultured on 5% sheep blood agar, and incubated for 24 h at 37 °C. DNA extraction was carried out with the QIAGEN body fluids extraction (column-based) kit. All isolates were screened for CTX-M 14 (Cefotaximase-Munich), CTX-M 15, NDM (New Delhi metallo-ß-lactamase), VIM (Verona Integron Encoded Metallo-Beta-Lactamase producing), IPM (imipenem), CMY (cephamycin), DHA (Docosahexaenoic acid), OXA-48 (Amber class D beta-lactamase), and KPC (Klebsiella pneumonia carbapenemase), using multiplex PCR, as described by Rotor-Gene real-time PCR (Qiagen Instrument AG: 5 Flex HRM) with the probe-based AMR multiplex PCR kit (Streck ARM-D kit beta-lactamases), via the previously described protocols by the manufacturers. Verification of the kit was conducted with positive and negative control cultures (*Escherichia coli* ATCC 25922, *Klebsiella pneumoniae* ATCC BAA 1705, *Klebsiella pneumoniae* ATCC BAA 1706, *Klebsiella pneumoniae* ATCC 70063, *Enterobacter cloaciae* NCTC 13406, *Klebsiella pneumoniae* ATCC BAA 2146).

##### ESBL and CTX-M Gene Identification

Molecular detection of antimicrobial resistant genes for all the isolates from humans (*n* = 200), poultry (*n* = 140), and aquaculture (*n* = 60) were carried out at the MRI. The represented samples were collected based on the cost of the kit and minimum required sample for interpretation. The isolates were sub-cultured on 5% sheep blood agar, and incubated for 24 h at 37 °C. DNA extraction was performed using the QIAGEN body fluids extraction (column-based) kit. The extracted DNA was screened for CTX-M 14, CTX-M 15, NDM, VIM, IPM, CMY, DHA, OXA-48, and KPC, using multiplex PCR, as described by Rotor-Gene real-time PCR (Qiagen Instrument AG: 5 Flex HRM) with the probe-based AMR multiplex PCR kit (Streck ARM-D kit beta-lactamases), as described by manufacturers’ protocols.

#### 4.2.2. AMU Investigation in the Non-Human Sector

This group prioritized studies in poultry and aquaculture sectors, as they are larger industries and growing sectors in Sri Lanka. This group excluded cattle farming, as it contributes to a minor industry in Sri Lanka.

##### AMU in Poultry Farms

Antimicrobial consumption (AMC) and antimicrobial usage (AMU) data were collected in the poultry sector in the selected two divisional secretary divisions (DSDs). Due to the limitations of travel during COVID-19 and budget limitations, the study was limited to two DSDs in the country. The calculation of AMU was referred to total antimicrobial used (mg) divided by total biomass of bird as population correction units (PCU) [37]. According to the European Surveillance of Veterinary Antimicrobial Consumption (ESVAC), one bird is considered 1 kg of mass (as standard weight). Thus, the total number of birds, multiplied by 1 kg, is the total weight in the calculation (www.ema.europa.eu; accessed on 20 January 2023).

AMC data for the research areas were collected from veterinary pharmaceutical product (VMP) distributors (*n* = 18), veterinary surgeons (*n* = 3), and feed manufacturers (*n* = 6), using specified formats. AMU information was collected from commercial broiler poultry farms (*n* = 56) in Kuliyapitiya and Panduwasnuwara DSDs, using a prepared questionnaire. It was based on sales or purchased antimicrobials from the suppliers, and national data that had not been included in the study. The highest density of poultry populations are found Kuliyapitiya and Panduwasnuwara in the country. In addition, the study was focused and limited to poultry as a study in two DSDs; information on other livestock species was not considered. The questionnaire was prepared and completed as investigator-administered type; farmers were interviewed in person by field officers and investigators visiting each farm physically. The survey was conducted between August and December, 2021. Furthermore, the dairy industry does not consume a significant amount of antimicrobials in the country due to the nature of farming, regulations and unavailability of appropriate preparation of antimicrobials in the country. Therefore, other livestock, including the cattle industry, were not included in the study herewith.

##### AMU in Aquaculture Farms

Antimicrobial use (AMU) data were collected in shrimp farms across North West Province, where 68 farms were surveyed. The aquaculture was limited to the coastal area of the province only (mainly in lagoons and closer to the sea); it was the reason to collect information from a province instead of a DSD. There was not a sufficient number of aquaculture farms in an individual DSD unit in the country. Paper-based, close-ended questionnaires were administered in person by investigators and field officers. Questions were asked regarding whether antimicrobials were used, and if they were used, the types of antimicrobials they were. Husbandry data were collected regarding water, stock and biosecurity management. Data from the field were transferred to an Excel spreadsheet. The survey was conducted between May and December 2021.

### 4.3. Database

For each farm and hospital, AMU and AMR data were extracted based on the above-mentioned survey and sample collection. AMU and AMR data were described clustered in terms of human, terrestrial, and aquatic food-animal production sectors on a shared database.

#### 4.3.1. AMR Data

The WHONET 2021 software was used for the recording of the ABST and gene data. The software was configured to suit the requirements of the health and animal sector. Data analysis was carried out using WHONET, according to the CLSI 2021 guidelines.

#### 4.3.2. AMC/U DATA

AMC data were collected on a master datasheet in Microsoft Excel format.

## 5. Conclusions

Antimicrobial resistance is alarming in each sector considered in the Kuliayapitiya and Paduwansnuwan divisional secretariats. High resistance was shown for commonly used antimicrobials in each sector, while ESBL and carbapenem resistance were identified as alarming. Antimicrobial usage needs to be minimized in the selected region, and a collective approach for AMR and AMU/AMC under the One Health concept is strongly encouraged.

## Figures and Tables

**Figure 1 antibiotics-12-00446-f001:**
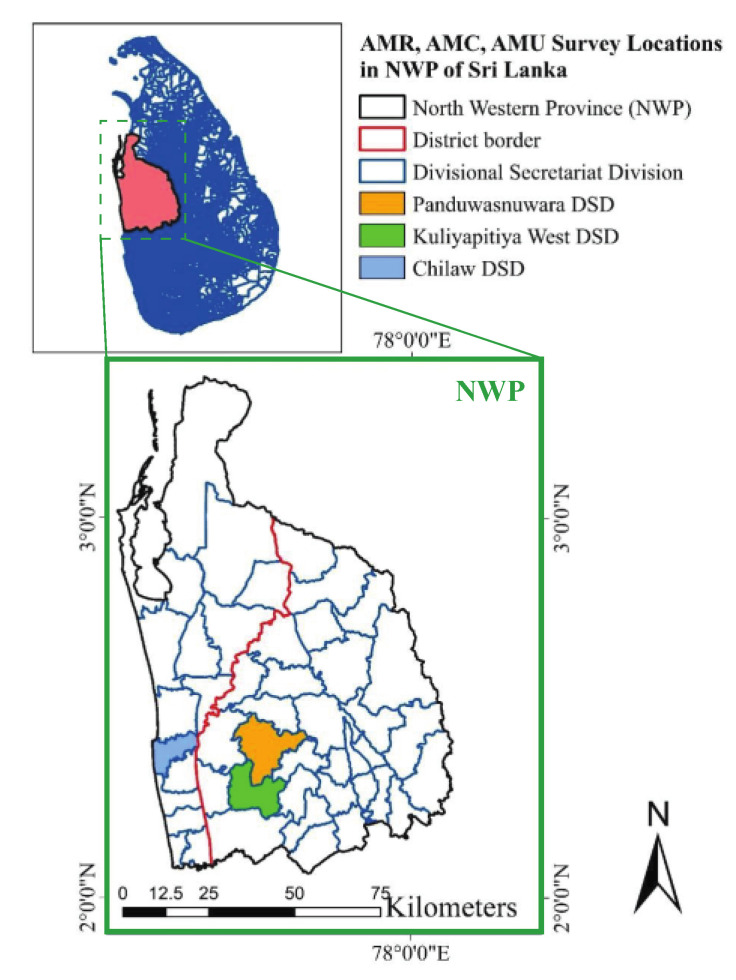
Three divisional secretariats, at Panduwasnuwara, Kuliyapitiya, and Chilaw, were established for the studies.

**Table 1 antibiotics-12-00446-t001:** Phenotypic antimicrobial resistance in *E. coli* isolates cultured from samples, collected from humans, poultry, and aquaculture in Paduwasnuwara and Kuliyapitiya divisional secretariats, Sri Lanka, 3 April 2021–30 November 2021.

		Human (Total *n* = 200)		Poultry (Total *n* = 401)		Aquaculture (Total *n* = 153)	
Antimicrobial Classes	Antimicrobials	Number of Resistant *E. coli*	Resistant %	Number of Resistant *E. coli*	Resistant %	Number of Resistant *E. coli*	Resistant %
Aminopenicillin	Ampicillin	133	66.5	379	94.5	62	40.5
Aminopenicillin-Beta-lactamase inhibitor	Amoxycillin-clavulanic acid	84	42	11	2.7	6	3.9
Cephalosporin	Ceftazidim	71	35.5	6	1.5	NT *	NT *
Cephalosporin	Cefotaxmine	71	35.5	30	7.5	NT *	NT *
Quinolones and Fluoroquinolones	Ciprofloxacin	70	35	103 ***	73.6	NT *	NT *
Chloramphenicol	Chloramphenicol	7	3.5	139	34.7	14	9.2
Quinolones and Fluoroquinolones	Enrofloxacin	NT *	NT *	137 **	68.5	NT *	NT *
Macrolide	Erythromycin	NT *	NT	393	98	138	90.2
Aminoglycoside	Gentamicin	25	12.5	97	27.2	69	45.1
Carbapenems	Imipenem	7	3.5	NT *	NT *	NT *	NT *
Carbapenems	Meropenem	8	4	5	1.3	19	12.4
Quinolones and Fluoroquinolones	Nalidixic acid	84	42	339	84.5	22	14.4
Tetracycline	Tetracycline	61	30.5	388	96.8	31	20.3
Sulfonamides	Trimethoprim-sulfamethoxazole	50	25	308	76.8	26	17
Sulfonamides	Trimethoprim	NT *	NT *	317	79.1	NT *	NT *

NT * Not tested, 103 *** tested out of 200 isolates, 137 ** tested out of 137 isolates.

**Table 2 antibiotics-12-00446-t002:** Heat map of MIC distribution of *E. coli* among humans, poultry, and aquaculture (Phoenix PX 2830).

Antimicrobials	0.094	0.125	0.19	0.25	0.38	0.5	0.75	1	1.5	2	3	4	6	8	12	16	24	32	48	64	96	128	256
** Human **																							
**Ampicillin**										3		56		3		3		131		4			
**Amoxicillin/Clavulonic acid**					1					4		77		29		5		81		3			
**Ceftazidime**								98				31						58		13			
**Cefotaxmin**								125		0		1	1					3		70			
**Cefoxitin**										1		170		7				20		2			
**Imipenem**								193				0				7							
**Meropenem**								193				0				7							
**Gentamicin**						9		81		77		4		3		25		1					
**Ciprofloxacin**		11		115		2		2		2		66											
**Trimethoprim/Sulfamethoxazole**				31		111		4		2		52											
**Chloramphenicol**								1		4		148		37		3		6		1			
**Tetracycline**								44		69		23				58		6					
** Poultry **																							
**Ampicillin**												14		1				125					
**Piperacillin**												19		8		5		7		25		51	
**amoxicillin/Clavulonic acid**												19		8		5		7		25		51	
**Ceftazidime**								1		15		38		61		24		1					
**Cefotaxmin**								24				115											
**Cefepime**										112		1		1		1							
**Cefoxitin**												136		3		1							
**Azetronam**										114		0						1					
**Imipenem**								112		1		2											
**Meropenem**								136				4											
**Amikacin**														115									
**Gentamicin**								24		102		11											
**Ciprofloxacin**				37		0		3		3		97											
**Trimethoprim/Sulfamethoxazole**						55		1		4		72		6		2							
**chloramphenicol**												94		23		3		20					
**Tetracycline**								3		7						122		8					
** Aquaculture **																							
**Ampicillin**												44		1		1		9					
**Piperacillin**												46		1		1						7	
**Amoxicillin/Clavulonic acid**								2		1		18		33		1							
**Ceftazidime**								55				0											
**Cefotaxmin**								3		0		51								1			
**Cefepime**										54								1					
**Cefoxitin**												55											
**Aztreonam**										54		0						1					
**Imipenem**								55				0											
**Meropenem**								55				0											
**Amikacin**														55									
**Gentamicin**						1		2		52													
**Ciprofloxacin**				54		0		1															
**Trimethoprim/Sulfamethoxazole**						47						8											
**Chloramphenicol**												39		14				2					
**Tetracycline**								3		49						3							

Red--Highest value; Yellow--50th Percentile; Orange--approximately 25th Precentile; Green--Lowest value 
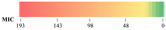
.

**Table 3 antibiotics-12-00446-t003:** Susceptibility/resistance of *E. coli* isolated from shrimps and *E. coli* isolated from shrimp culture pond water.

Antimicrobials	*E. coli* from Shrimp (*n* = 60)			*E. coli* from Shrimp Pond Water (*n* = 93)		
	R%	I%	S%	R%	I%	S%
Amoxicillin/Clavulanic acid	1.67	16.67	81.66	5.38	22.58	72.04
Ampicillin	28.33	10	61.67	48.39	9.68	41.94
Amoxicillin	0	0	0	0	0	0
Chloramphenicol	1.67	8.33	90	13.98	6.45	79.57
Erythromycin	81.67	18.33	0	95.7	4.3	0
Gentamicin	45	26.67	28.33	54.16	36.56	18.28
Meropenem	10	20	70	13.98	35.48	50.54
Nalidixic acid	16.67	30	53.33	12.9	45.16	41.94
Neomycin	0	0	0	0	0	0
Oxytetracycline	0	0	0	0	0	0
Trimethoprim/Sulfamethoxazole	16.67	1.67	81.66	22.58	4.3	73.12
Tetracycline	11.67	0	88.33	20.43	2.15	77.42

**Table 4 antibiotics-12-00446-t004:** Resistant genes identified in the selected isolates from humans, poultry, and aquaculture (*n* = 200).

Resistant Gene	Number of Positive Isolates	%
Carbapenem	6	3
AmpC	20	10
ESBL	71	35.5
CTX-15	51	25.5
CTX-14	7	3.5
OXA-48	1	0.5
CTX-M and OXA-48	134	67
MHT/Modified Hodge test positive	8	4
MCIM/ Carbapenem inactivation method	6	3
Total isolates	304	152

**Table 5 antibiotics-12-00446-t005:** Sector, sample types, sites, and number of samples collected from each sector during the study for AMR and AMC/AMU.

Sites	Sectors	Study Focus	Type of Sample	Sample Number and Unit
Kuliyapitiya and Chillaw	Human	AMR	Urine	200 individuals
Paduwasnuwara, Kuliyapitiya	Poultry	AMR	Cloaca	401 broilers
Chillaw	Aquaculture	AMR	Shrimp & pond water	153 shrimps
Paduwasnuwara, Kuliyapitiya	Poultry	AMC/AMU	Data	18 veterinary Pharmaceutical product distributors
Paduwasnuwara, Kuliyapitiya	Poultry	AMC/AMU	Data	56 poultry farms
Paduwasnuwara, Kuliyapitiya	Poultry	AMC/AMU	Data	3 government veterinary surgeons
Paduwasnuwara, Kuliyapitiya	Poultry	AMC/AMU	Data	6 feed manufacturers
Chilaw	Aquaculture	AMC/AMU	Data	68 shrimp farms
Chilaw	Aquaculture	AMC/AMU	Data	17 shrimp hatcheries

## Data Availability

Not applicable.

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
