# Peer review of "One Health Surveillance of Antimicrobial Use and Resistance: Challenges and Successes of Implementing Surveillance Programs in Sri Lanka"

_antibiotics, 2023, doi:10.3390/antibiotics12030446_

Round 1

Reviewer 1 Report

The manuscript add data in the knowledge of the complex puzzle of the antimicrobial resistance in Sri Lanka. However, in its present form suffer by several technical mistakes. I support its possible further processing after the clarification of the mentioned issues below:

L36: “we collected” – please avoid the using of personal verb mode formulations, it is not so characteristic for the scientific style. Please revise this throughout the manuscript!

L44: “was highest” instead of “is highest

L54: being the first appearance in the text must become “antimicrobial resistance (AMR)

L54: please provide citations in rectangular brackets as number, according to the journal requirement

L70, L78 – being the Introduction section, it would be important to insert documented references at the end of each paragraph

L92-94: these sentences are more suitable to the materials and methods section, rather than results…

L101: The presented tables 1-5 design are not in agreement with the journal requirement

L174: the Table 3 must be in a landscape page. Also, it would be important to indicate the class of the tested antimicrobials (especially in Table 2)

L333 – please provide appropriate subheading numbers

L334: “E. coli” – the authors must ensure that all of the scientific name of species are written in italics. Please carefully revise this concern throughout the manuscript.

L336: “Tryptose Soy broth” – please mention after brackets in case of each of the used reagents and equipment the production company name city and country Please carefully revise this concern throughout the manuscript.

L353: “370C,” – please revise throughout the manuscript!

L357: can you provide any reason why only one E. coli isolate was selected for further processing

L366: must become “water (n= 93) and shrimp (n=60) origin E. coli samples ...”

L400: please specify the meaning of „CTX-M 14, CTX-M 15, NDM, VIM, IPM, CMY, 400 DHA, OXA-48”

L401: within each of the PCR reaction the authors must document the using of positive and negative controls

L403: please include and indicate the manusfacturer protocol as reference at the end of the paragraph

L439: within the conclusion section the authors must evidence the study limitations and future perspectives in the approached research area

L468: THE REFERENCE LIST IN THE PRESENT FORM IS NOT IN AGREEMNT WITH HE JOURNAL REQUIREMENT. Please carefully revise this concern!

Overall, I beleive that the reference list must to be substantially improved, reaching the value of 30, as mentioned in the guideline for the authors. In this regards, and especially within the introduction section, a clear definition of pathogenic E. coli is needed, mentioning the classification of pathotypes, in order to highlight the study importance. Also, the authors must highlight the importance of E. coli as foodborne pathogen, which can be easily transmitted to the consumer via animal origin foodstuffs (e.g. cheese - doi: 10.3390/antibiotics11060721 or beef - doi: 10.3390/foods9111543). This articles can be consulted and cited, providing important data in E. coli AMR resistance.

Author Response

Reviewer 1

The manuscript add data in the knowledge of the complex puzzle of the antimicrobial resistance in Sri Lanka. However, in its present form suffer by several technical mistakes. I support its possible further processing after the clarification of the mentioned issues below:

  1. L36: “we collected” – please avoid the using of personal verb mode formulations, it is not so characteristic for the scientific style. Please revise this throughout the manuscript!

Response 1: Thank you for the comment. “We collected” was changed to “the Fleming Fellows collected,…” in line 36.

  1. L44: “was highest” instead of “is highest”

Response 2: Thank you for the recommendation. Tenses are changed from present to past tenses where appropriate particular “was” in lines 44 and 45.

  1. L54: being the first appearance in the text must become “antimicrobial resistance (AMR)”

Response 3: Thank you for the advice. Abbreviation is added in line 55.

  1. L54: please provide citations in rectangular brackets as number, according to the journal requirement

Response 4: Thank you for the reminder. Citations are changed to brackets as numbers.

  1. L70, L78 – being the Introduction section, it would be important to insert documented references at the end of each paragraph

Response 5: Thank you for the thoughtful recommendation. Relevant writings with references are added to lines 71-72 and 83-86.

  1. L92-94: these sentences are more suitable to the materials and methods section, rather than results…

Response 6: The sentences, figure and table are moved to Materials and Methods section.

  1. L101: The presented tables 1-5 design are not in agreement with the journal requirement

Response 7: Thank you for pointing out the table design. It is true they are not ideal when copy-pasted in the unformatted editable version. Please find PDF files in place of previous diagrams and tables. We have also attached a supplementary word file (All diagrams) which contains diagrams and tables in PDF accompanied with formatted text format.

  1. L174: the Table 3 must be in a landscape page. Also, it would be important to indicate the class of the tested antimicrobials (especially in Table 2)

Response 8: Thank you for the recommendation. Table 3 is revised as heat map of MIC distribution. Hopefully this presents data better.  

Class of antimicrobials is added to first column in Table 2.

  1. L333 – please provide appropriate subheading numbers

Response 9: Sub-heading numbers are revised.

  1. L334: “ coli” – the authors must ensure that all of the scientific name of species are written in italics. Please carefully revise this concern throughout the manuscript.

Response 10: E. Coli is italicized.

  1. L336: “Tryptose Soy broth” – please mention after brackets in case of each of the used reagents and equipment the production company name city and country Please carefully revise this concern throughout the manuscript.

Response 11: Please find production company name in brackets throughout manuscript.

  1. L353: “370C,” – please revise throughout the manuscript!

Response 12: Our apology for the un-superscripted font.

  1. L357: can you provide any reason why only one  coliisolate was selected for further processing

Response 13: Thank you for reviewer’s comment. There is only limited economic resources in this study so only one indicator bacteria is applied. Also as inspired by comments and indicated in introduction, E. coli is prevalent in human and animals, has become significant in One Health, food chain, increasing resistance, and its culture is relatively feasible in the laboratories.

  1. L366: must become “water (n= 93) and shrimp (n=60) origin  colisamples ...”

Response 14: Revised, thank you for comment.

  1. L400: please specify the meaning of „CTX-M 14, CTX-M 15, NDM, VIM, IPM, CMY, 400 DHA, OXA-48”

Response 15: Thank you for comment. Meanings are specified in manuscript as CTX-M 14(Chefotaximase-Munich), CTX-M 15, NDM (New Delhi metallo-ß-lactamase), VIM (Verona Intergron Encoded Metallo-Beta-Lactamase producing), IPM(imipenem), CMY(cephamycin), DHA(trimethoprim), OXA-48, KPC(Klebsiella pneumonia carbapenemase)

  1. L401: within each of the PCR reaction the authors must document the using of positive and negative controls

Response 16: Thank you for comment. Multiplex PCR as described by Rotor Gene real time PCR (Qiagen Instrument AG: 5 Flex HRM) with probe based AMR multiplex PCR kit (Streck ARM-D kit Beta lactamases) as described protocol previously by the manufacturers.

  1. L403: please include and indicate the manusfacturer protocol as reference at the end of the paragraph

Response 17: All isolates were screened for CTX-M 14, CTX-M 15, NDM, VIM, IPM, CMY, DHA, OXA-48, KPC by multiplex PCR as described by Rotor Gene real time PCR (Qiagen Instrument AG: 5 Flex HRM) with probe based AMR multiplex PCR kit (Streck ARM-D kit Beta lactamases) as described protocol by the manufacturers.

  1. L439: within the conclusion section the authors must evidence the study limitations and future perspectives in the approached research area

Response 18: Thank you for the comment. It is indeed important to state the limitation. Writing is included in lines 218 to 231.

  1. L468: THE REFERENCE LIST IN THE PRESENT FORM IS NOT IN AGREEMNT WITH HE JOURNAL REQUIREMENT. Please carefully revise this concern!

Response 19: Thank you for the comment. The reference list is revised.

  1. Overall, I beleive that the reference list must to be substantially improved, reaching the value of 30, as mentioned in the guideline for the authors.

Response 20: Thank you for the comment. Cited writing (Lines 85-89, 237-238, 245-248) are added and the reference list is expanded.

  1. In this regards, and especially within the introduction section, a clear definition of pathogenic  coliis needed, mentioning the classification of pathotypes, in order to highlight the study importance.

Response 21: Thank you for the comment. It is indeed important to do so. Definition and importance to study E. coli is described in introduction.

  1. Also, the authors must highlight the importance of  colias foodborne pathogen, which can be easily transmitted to the consumer via animal origin foodstuffs (e.g. cheese - doi: 10.3390/antibiotics11060721 or beef - doi: 10.3390/foods9111543). This articles can be consulted and cited, providing important data in E. coli AMR resistance.

Response 22: Thank you for the input. The reference and the importance of E. coli is described in introduction.

Reviewer 2

The authors of this article investigate the One Health surveillance of antimicrobial resistance and antimicrobial use in Sri Lanka. Authors use questionnaires and survey techniques to examine the pattern of AMU, while AST and genomics were used to investigate AMR. According to the authors, beta-lactam resistance is common in humans, and erythromycin resistance is common in non-humans. This is a highly desirable study in Sri Lanka.

However, the study has some serious issues in the methodology section, which should be addressed.

  1. The AMU section's authors did not explain how they determined sample size or why they collected data from only two of the country's 331 divisional secretary divisions.

Response 23:

--Is this by convenience sampling?

Yes it is partly by convenience sampling and sampling approach is clarified in lines 334-335. But there is consideration for the locality and geographic characteristics. Author write in lines 489-492 depicted “The aquaculture was limited to the coastal area of the province only (mainly in lagoons and closer to the sea); it was the reason to collect information from a province instead of a DSD. There was no sufficient number of aquaculture farms in an individual DSD unit in the country.”

In poultry AMU study, in lines 477 and 478 describes “Kuliyapitiya and Panduwasnuwara are where the highest density of poultry population are found in the country.”

--Any COVID-19 imposed difficulty?

Thank you reviewer for this question. The Fellows who studied poultry and aquaculture experienced significant difficulty due to COVID-19’s travel restriction and meeting restrictions mentioned in lines 462 to 463. It was difficult to find driver and security staff to accompany the field visits (necessary part of the field study). “Due to limitations of travel during COVID-19 and budget limitations, the study was limited to two DSDs in the country”. As overnight rest area was closed, it was impossible to travel to places further then day trips.

  1. How (basis) do they prepare the questionnaire, what type of data do they collect, and how do they calculate AMU?

Response 24: Questionnaire is paper-based, investigator-administered closed ended questions.

AMU is inquired by type in aquaculture assessment (lines 490-493). Poultry AMU by quantity (lines 128-130), categorical data identified from local farm shops (line 133-137) and conversation with poultry farmers (informal narrative).

  1. Why did they only collect poultry AMU data from two divisional secretary's divisions (DSD) while aquaculture data was collected from the entire province? Why they did not collect any data from cattle, etc.?

Response 25: Thank you for the comment. Poultry from DSD and aquaculture sites choice rationale is described in Response 23. Cattle is not studied according to the small size and limited antimicrobial use of the industry (lines 477-483).

  1. What were the formulas to calculate the AMU? 

-Response 26: Thank you for the question. Tylosin was calculated by dividing active substance in mg by estimated biomass of population correction unit (mg/PCU) in line 107-108.

  1. Furthermore, the English were found to be poor throughout the manuscript, requiring extensive editing.

Response 27: Thank you for the comment. English editing service is engaged and completed through MDPI.

General comments

  1. The methodology is not scientific which required many unavoidable changes. The quality of the figures is poor, and the authors did not explain how did (the tools) they draw. The results are extremely confusing; The tables are inadequately formatted; even I can't see the entire table 3.

Response 28: Thank you for the comment. The figures and tables are revised as PDF file with formatted editable text (RTF). A supplementary document with tables and figures is included.

  1. The author should separately write an interpretation of the AMU, AMC, and AMR.

They can draw a comparison between the distributed and used antimicrobials. Propers headings should be made for each AMC, AMU and AMR, such as AMU in poultry, AMU in Humans etc.

Response 29: Thank you for the comment. This is a valid point. Results are written under headings of:

2.1 Overview of AMU and AMR in non-human sector

2.2 Antimicrobial use in farms—by wholesale, by retail, by use/consumption, and by prescription.

2.3 Antimicrobial resistance in human and non-human sector (poultry and aquaculture) and environment

Minor comments

  1. Line 54: Why authors did not denote antimicrobial resistance (AMR) in the first sentence of the introduction while they did in the second sentence?

Response 30: Thank you for the comment. It is a mistake. AMR is denoted.

  1. In line 54 author should cite DOI: 10.29261/pakvetj/2021.067a http://www.pvj.com.pk/pdf-files/41_4/519-523.pdf

Response 31: Thank you for the comment. The citation is added.

  1. Lines 57 and 58: How AMR is responsible for poverty, explain and cite the world bank report. It is better to cite the specific report of the World Bank.

Response 31: Thank you for the comment. The citation of report from World Bank is cited.

  1. Line 59, 60, 61, and 62: The sentence is very confusing and lengthy, please rephrase the sentence.

Response 33: English editing was engaged and sentence edited.

  1. Line 63: What is the relevance of the Scientific scholarship with this study, what mean to say, I think the sentence is not needed please remove it.

Response 34: Thank you for the comment. It is removed according to recommendation.

  1. In line, 66 authors should cite DOI: http://www.pvj.com.pk/abstract/42_3/22-010.htm

Response 35: Citation is added.

  1. In lines, 88 and 89 authors wrote the article describing the cross-sectional data of AMU and AMU. How they calculated AMR cross-sectionally?

Response 36: Thank you for the comment. Cross-sectional meant point-prevalence of the AMR percentage.

  1. Data presented between lines 92 and 107 should be part of the methodology section.

Response 37: Thank you for the comment, please find lines 92-107 moved to methodology section.

  1. In lines 108 and 109, the authors calculated the biomass of poultry, but how they did it is not clear in the methodology section.

Response 38: Thank you for the comment. The calculation is described in lines 466 to 470 in revised version “PCU pertained to the total number of birds surveyed multiplied by the ESVAC (European Surveillance of Veterinary Antimicrobial Consumption) standard weight at treatment for a broiler chicken. According to the ESVAC, one bird is considered as 1 kg of biomass, and 1 kg was multiplied by a number of birds in the given population (www.ema.europa.eu).”

  1. Line 334: How do authors know they are collecting  coli while in line 337 they are saying confirmation was done at MRI? 

Response 39: Thank you for the opportunity to write clearer. Lines 335 to 338 and 342 to 345 are written to clarify the identification and reconfirmation process.

Suggestions:

  1. Please start the discussion in One Health Perspectives of AMU and AMR and cite the articles doi: 10.3389/fpubh.2021.693703

Response 40: Thank you for the suggestion. We added the recommended article where One Health perspective is discussed.

  1. Authors need to follow the manuscripts https://doi.org/10.3390/antibiotics10050598and doi: 10.3389/fvets.2021.673809 for AMU formulas.

Response 41: Thank you for the insightful input regarding AMU formulas. Citation and writing is added in line 108 (https://doi.org/10.3390/antibiotics10050598).

Additional citation is included in line 468 methodology (10.3389/fvets.2021.673809)

Reviewer 2 Report

The authors of this article investigate the One Health surveillance of antimicrobial resistance and antimicrobial use in Sri Lanka. Authors use questionnaires and survey techniques to examine the pattern of AMU, while AST and genomics were used to investigate AMR. According to the authors, beta-lactam resistance is common in humans, and erythromycin resistance is common in non-humans. This is a highly desirable study in Sri Lanka.

However, the study has some serious issues in the methodology section, which should be addressed. The AMU section's authors did not explain how they determined sample size or why they collected data from only two of the country's 331 divisional secretary divisions. How (basis) do they prepare the questionnaire, what type of data do they collect, and how do they calculate AMU? Why did they only collect poultry AMU data from two divisional secretary's divisions (DSD) while aquaculture data was collected from the entire province? Why they did not collect any data from cattle, etc.? What were the formulas to calculate the AMU? Furthermore, the English were found to be poor throughout the manuscript, requiring extensive editing.

General comments

The methodology is not scientific which required many unavoidable changes. The quality of the figures is poor, and the authors did not explain how did (the tools) they draw. The results are extremely confusing; The tables are inadequately formatted; even I can't see the entire table 3. The author should separately write an interpretation of the AMU, AMC, and AMR. They can draw a comparison between the distributed and used antimicrobials. Propers headings should be made for each AMC, AMU and AMR, such as AMU in poultry, AMU in Humans etc.

Minor comments

Line 54: Why authors did not denote antimicrobial resistance (AMR) in the first sentence of the introduction while they did in the second sentence?

In line 54 author should cite DOI: 10.29261/pakvetj/2021.067a http://www.pvj.com.pk/pdf-files/41_4/519-523.pdf

Lines 57 and 58: How AMR is responsible for poverty, explain and cite the world bank report. It is better to cite the specific report of the World Bank.

Line 59, 60, 61, and 62: The sentence is very confusing and lengthy, please rephrase the sentence.

Line 63: What is the relevance of the Scientific scholarship with this study, what mean to say, I think the sentence is not needed please remove it.

In line, 66 authors should cite DOI: http://www.pvj.com.pk/abstract/42_3/22-010.htm

In lines, 88 and 89 authors wrote the article describing the cross-sectional data of AMU and AMU. How they calculated AMR cross-sectionally?

Data presented between lines 92 and 107 should be part of the methodology section.

In lines 108 and 109, the authors calculated the biomass of poultry, but how they did it is not clear in the methodology section.

Line 334: How do authors know they are collecting E. coli while in line 337 they are saying confirmation was done at MRI? 

Suggestions:

Please start the discussion in One Health Perspectives of AMU and AMR and cite the articles doi: 10.3389/fpubh.2021.693703

Authors need to follow the manuscripts https://doi.org/10.3390/antibiotics10050598 and doi: 10.3389/fvets.2021.673809 for AMU formulas.

Author Response

Reviewer 2

The authors of this article investigate the One Health surveillance of antimicrobial resistance and antimicrobial use in Sri Lanka. Authors use questionnaires and survey techniques to examine the pattern of AMU, while AST and genomics were used to investigate AMR. According to the authors, beta-lactam resistance is common in humans, and erythromycin resistance is common in non-humans. This is a highly desirable study in Sri Lanka.

However, the study has some serious issues in the methodology section, which should be addressed.

  1. The AMU section's authors did not explain how they determined sample size or why they collected data from only two of the country's 331 divisional secretary divisions.

Response 1:

--Is this by convenience sampling?

Yes it is partly by convenience sampling and sampling approach is clarified in lines 334-335. But there is consideration for the locality and geographic characteristics. Author write in lines 489-492 depicted “The aquaculture was limited to the coastal area of the province only (mainly in lagoons and closer to the sea); it was the reason to collect information from a province instead of a DSD. There was no sufficient number of aquaculture farms in an individual DSD unit in the country.”

In poultry AMU study, in lines 477 and 478 describes “Kuliyapitiya and Panduwasnuwara are where the highest density of poultry population are found in the country.”

--Any COVID-19 imposed difficulty?

Thank you reviewer for this question. The Fellows who studied poultry and aquaculture experienced significant difficulty due to COVID-19’s travel restriction and meeting restrictions mentioned in lines 462 to 463. It was difficult to find driver and security staff to accompany the field visits (necessary part of the field study). “Due to limitations of travel during COVID-19 and budget limitations, the study was limited to two DSDs in the country”. As overnight rest area was closed, it was impossible to travel to places further then day trips.

  1. How (basis) do they prepare the questionnaire, what type of data do they collect, and how do they calculate AMU?

Response 2: Questionnaire is paper-based, investigator-administered closed ended questions.

AMU is inquired by type in aquaculture assessment (lines 490-493). Poultry AMU by quantity (lines 128-130), categorical data identified from local farm shops (line 133-137) and conversation with poultry farmers (informal narrative).

  1. Why did they only collect poultry AMU data from two divisional secretary's divisions (DSD) while aquaculture data was collected from the entire province? Why they did not collect any data from cattle, etc.?

Response 3: Thank you for the comment. Poultry from DSD and aquaculture sites choice rationale is described in Response 23. Cattle is not studied according to the small size and limited antimicrobial use of the industry (lines 477-483).

  1. What were the formulas to calculate the AMU? 

-Response 4: Thank you for the question. Tylosin was calculated by dividing active substance in mg by estimated biomass of population correction unit (mg/PCU) in line 107-108.

  1. Furthermore, the English were found to be poor throughout the manuscript, requiring extensive editing.

Response 5: Thank you for the comment. English editing service is engaged and completed through MDPI.

General comments

  1. The methodology is not scientific which required many unavoidable changes. The quality of the figures is poor, and the authors did not explain how did (the tools) they draw. The results are extremely confusing; The tables are inadequately formatted; even I can't see the entire table 3.

Response 6: Thank you for the comment. The figures and tables are revised as PDF file with formatted editable text (RTF). A supplementary document with tables and figures is included.

  1. The author should separately write an interpretation of the AMU, AMC, and AMR.

They can draw a comparison between the distributed and used antimicrobials. Propers headings should be made for each AMC, AMU and AMR, such as AMU in poultry, AMU in Humans etc.

Response 7: Thank you for the comment. This is a valid point. Results are written under headings of:

2.1 Overview of AMU and AMR in non-human sector

2.2 Antimicrobial use in farms—by wholesale, by retail, by use/consumption, and by prescription.

2.3 Antimicrobial resistance in human and non-human sector (poultry and aquaculture) and environment

Minor comments

  1. Line 54: Why authors did not denote antimicrobial resistance (AMR) in the first sentence of the introduction while they did in the second sentence?

Response 8: Thank you for the comment. It is a mistake. AMR is denoted.

  1. In line 54 author should cite DOI: 10.29261/pakvetj/2021.067a http://www.pvj.com.pk/pdf-files/41_4/519-523.pdf

Response 9: Thank you for the comment. The citation is added.

  1. Lines 57 and 58: How AMR is responsible for poverty, explain and cite the world bank report. It is better to cite the specific report of the World Bank.

Response 10: Thank you for the comment. The citation of report from World Bank is cited.

  1. Line 59, 60, 61, and 62: The sentence is very confusing and lengthy, please rephrase the sentence.

Response 11: English editing was engaged and sentence edited.

  1. Line 63: What is the relevance of the Scientific scholarship with this study, what mean to say, I think the sentence is not needed please remove it.

Response 12: Thank you for the comment. It is removed according to recommendation.

  1. In line, 66 authors should cite DOI: http://www.pvj.com.pk/abstract/42_3/22-010.htm

Response 13: Citation is added.

  1. In lines, 88 and 89 authors wrote the article describing the cross-sectional data of AMU and AMU. How they calculated AMR cross-sectionally?

Response 14: Thank you for the comment. Cross-sectional meant point-prevalence of the AMR percentage.

  1. Data presented between lines 92 and 107 should be part of the methodology section.

Response 15: Thank you for the comment, please find lines 92-107 moved to methodology section.

  1. In lines 108 and 109, the authors calculated the biomass of poultry, but how they did it is not clear in the methodology section.

Response 16: Thank you for the comment. The calculation is described in lines 466 to 470 in revised version “PCU pertained to the total number of birds surveyed multiplied by the ESVAC (European Surveillance of Veterinary Antimicrobial Consumption) standard weight at treatment for a broiler chicken. According to the ESVAC, one bird is considered as 1 kg of biomass, and 1 kg was multiplied by a number of birds in the given population (www.ema.europa.eu).”

  1. Line 334: How do authors know they are collecting  coli while in line 337 they are saying confirmation was done at MRI? 

Response 17: Thank you for the opportunity to write clearer. Lines 335 to 338 and 342 to 345 are written to clarify the identification and reconfirmation process.

Suggestions:

  1. Please start the discussion in One Health Perspectives of AMU and AMR and cite the articles doi: 10.3389/fpubh.2021.693703

Response 18: Thank you for the suggestion. We added the recommended article where One Health perspective is discussed.

  1. Authors need to follow the manuscripts https://doi.org/10.3390/antibiotics10050598and doi: 10.3389/fvets.2021.673809 for AMU formulas.

Response 19: Thank you for the insightful input regarding AMU formulas. Citation and writing is added in line 108 (https://doi.org/10.3390/antibiotics10050598).

Additional citation is included in line 468 methodology (10.3389/fvets.2021.673809)

Round 2

Reviewer 1 Report

The authors correctly acknowledged all of the raised concerns.

Congratulations!

Reviewer 2 Report

My only concern is the quality of figure 1.